

# Air quality improvement during triple-lockdown in the coastal city of Kannur, Kerala to combat Covid-19 transmission

C.T. Resmi[1], T. Nishanth[2], M.K. Satheesh Kumar[3], M.G. Manoj[4], M. Balachandramohan[1] and K.T. Valsaraj[5]

[1] Department of Physics, Erode Arts and Science College, Erode, Tamil Nadu, India
[2] Department of Physics, Sree Krishna College Guruvayur, Thrissur, Kerala, India
[3] Department of Atomic and Molecular Physics, Manipal Academy of Higher Education, Manipal, Karnataka, India
[4] Advanced Centre for Atmospheric Radar Research, Cochin University of Science and Technology, Cochin, Kerala, India
[5] Cain Department of Chemical Engineering, Louisiana State University, Baton Rouge, LA, USA

Corresponding author
T. Nishanth, nisthu.t@gmail.com

## ABSTRACT

The novel SARS-CoV-2 coronavirus that emerged in the city of Wuhan, China, last year has since become the COVID-19 pandemic across all continents. To restrict the spread of the virus pandemic, the Government of India imposed a lockdown from 25 March 2020. In India, Kannur district was identified as the first "hotspot" of virus transmission and a "triple-lockdown" was implemented for a span of twenty days from 20 April 2020. This article highlights the variations of surface $O_3$, NO, $NO_2$, CO, $SO_2$, $NH_3$, VOC's, $PM_{10}$, $PM_{2.5}$ and meteorological parameters at the time of pre-lockdown, lockdown and triple-lockdown days at Kannur town in south India using ground-based analyzers. From pre-lockdown days to triple-lockdown days, surface $O_3$ concentration was found to increase by 22% in this VOC limited environment. NO and $NO_2$ concentrations were decreased by 61% and 71% respectively. The concentration of $PM_{10}$ and $PM_{2.5}$ were observed to decline significantly by 61% and 53% respectively. Reduction in $PM_{10}$ during lockdown and triple-lockdown days enhanced the intensity of solar radiation reaching the lower troposphere, and increased air temperature and reduced the relative humidity. Owing to this, surface $O_3$ production over Kannur was found to have increased during triple-lockdown days. The concentration of CO (67%), VOCs (61%), $SO_2$ (62%) and $NH_3$ (16%) were found to decrease significantly from pre-lockdown days to triple-lockdown days. The air quality index revealed that the air quality at the observational site was clean during the lockdown.

## INTRODUCTION

Air pollution is a major environmental issue that affects people in developed and developing countries alike. Particulate matters ($PM_{10}$ and $PM_{2.5}$), oxides of nitrogen (NO and $NO_2$), sulfur dioxide ($SO_2$), ozone ($O_3$), carbon monoxide (CO) and volatile organic

compounds (VOC's) are the most common atmospheric air pollutants encountered in our daily life (*Chen & Kan, 2008*; *Guo et al., 2019*). At the ground level, $O_3$ is a major secondary air pollutant and greenhouse gas, produced from its precursor gases in the presence of solar radiation, and it plays a crucial role in air quality (*Yadav et al., 2016*; *Lu et al., 2018*; *Ding et al., 2020*; *Maji, Beig & Yadav, 2020*; *Resmi et al., 2020*). Particulate matters are complex mixture of organic and inorganic substances found in the ambient air, and they play a vital role in the radiation budget of the atmosphere via the scattering and absorption processes (*Qu et al., 2017*, *2018*). The major anthropogenic sources of $PM_{10}$ and $PM_{2.5}$ are vehicular emission, industry, building construction, quarrying and mining cement plants, ceramic industries and burning of fossil fuel power plants (*Cheng et al., 2006*; *Liu et al., 2015*). *Guo et al. (2017)* revealed that, in India more than one million people died in 2015 due to particulate matter pollution.

Carbon monoxide is an important trace pollutant that influences the oxidizing capacity of the atmosphere, and the concentration of the surface $O_3$ by removing hydroxyl radicals, the primary oxidant in the troposphere (*Duncan & Logan, 2008*; *Yadav et al., 2019a*). The main CO sources in the atmosphere are anthropogenic and natural (*Lawrence & Lelieveld, 2010*). Inhalation of CO is considered very toxic to humans because it can cause acute intoxication (*Kinoshita et al., 2020*). VOC's emitted in the atmosphere consist of saturated and unsaturated hydrocarbons, aromatic hydrocarbons, and halogenated organic compounds (*Liu et al., 2008*; *Montero-Montoya, Lopez-Vargas & Arellano-Aguilar, 2018*). These are emitted into the atmosphere by a number of industrial activities such as petrochemical process, storage, distribution, paint, solvent, combustion processes and motor vehicle exhaust (*Franco et al., 2015*; *Yadav et al., 2019b*). Isoprene and monoterpenes are another reactive natural VOC's commonly found in the lower atmosphere, which are mainly emitted by biogenic sources like trees and plants (*Fuentes et al., 2000*; *Menchaca-Torre, Mercado-Hernandez & Mendoza-Domínguez, 2015*). These hydrocarbons play a crucial role in the photochemical production of $O_3$ and other oxidants in the lower atmosphere (*Atkinson, 2000*; *Srivastava, Sengupta & Dutta, 2005*).

Sulfur dioxide is released into the atmosphere through both natural and anthropogenic emissions. Natural sources are mainly by volcanic eruptions while anthropogenic sources include the combustion of all sulfur containing fuels like oil coal and diesel used for the power generation for industrial activities (*Mallik & Lal, 2014*; *Zhang et al., 2017*). Industrial and traffic emissions are the major ammonia ($NH_3$) sources in an urban environment (*Pandolfi et al., 2012*; *Phan et al., 2013*).

Severe Acute Respiratory Syndrome Corona Virus-2 (SARS-CoV2) is the pathogenic agent of Covid-19, a disease first reported from Wuhan Hubei Province of China in December 2019. It was declared a global pandemic by the World Health Organization (WHO) on 11 March 2020 (*Al-Qahtani, 2020*). The course of the disease is often mild undistinguishable from severe pneumonia which eventually lead to acute respiratory distress syndrome (ARDS) and death (*Lu, Stratton & Tang, 2020*). India reported the first confirmed case of coronavirus infection on 30 January 2020 in the southern state of Kerala (*Gautam & Hens, 2020*). To stop the spread of the virus pandemic, many countries have decided to enforce lockdown measures, even as they caused a severely downturn on the

global economy. Subsequently, the Government of India imposed a countrywide lockdown to implement the *Break-the-Chain* mission to curtail its spread for 21 days in its first phase, and extended up to a second spell of 19 days and a final third phase of 14 days up to May 17 for a total of 54 days. As a result, tightened restrictive measures (i.e., closure all academic institutes, industries, markets, malls and all public places, non-essential businesses, limitation of motorized transports, shut down of Indian railway network, cancelation of inbound and outbound flights) were implemented throughout the nation to impose social distancing.

A substantial enhancement in the air quality in the lockdown period from all over the world were reported. Air pollution in China was significantly reduced as more people quarantined to prevent the social spread of Covid-19 (*Dutheil, Baker & Navel, 2020*; *Muhammad, Long & Salman, 2020*; *Wang & Su, 2020*). *Bao & Zhang (2020)* reported that the concentrations of $SO_2$, $PM_{2.5}$, $PM_{10}$, $NO_2$ and CO over 44 cities in northern China have decreased significantly by 6.76%, 5.93%, 13.66%, 24.67% and 4.58%, respectively due to vehicular restrictions during lockdown period. In addition to these, they conclude that these reduction in air pollutants caused a decrease in air quality index (AQI) by 7.80% over these cities in China. Likewise, Community Multi-scale Air Quality Model (CMAQ) analysis carried out by *Wang et al. (2020)* found that, $PM_{2.5}$ concentration over different parts of urban areas in China were significantly reduced in lockdown period. The findings of *Shi & Brasseur (2020)* showed that particulate matter pollution decreased by an average of 35 percent and nitrogen dioxide decreased by an average of 60 percent in northern China after the lockdowns began on January 23. However, the study found the average surface $O_3$ concentration increased by a factor of 1.5–2 over the same time period. *Bauwens et al. (2020)* also observed a substantial decrease of $NO_2$ column by 40% over cities in China and 20–38% in Western Europe and USA.

*Kanniah et al. (2020)* reported that $PM_{10}$, $PM_{2.5}$, $NO_2$, $SO_2$ and CO concentrations were reduced to 26–31%, 23–32%, 63–64%, 9–20% and 25–31%, respectively in urban regions of Malaysia during the lockdown period. Further, they revealed that the restricted industrial activities imposed in lockdown period resulted a reduced concentrations of Aerosol Optical Depth and tropospheric $NO_2$ over East Asian region. There were reports that air pollution was significantly reduced in Barcelona in Spain (*Tobías et al., 2020*) and European region (*Sicard et al., 2020*) due to the lockdown to prevent Covid-19 spread. A decline in $NO_2$, NO and an increase in surface $O_3$ was observed at Sao Paulo in Brazil during lockdown period (*Dantas et al., 2020*; *Nakada & Urban, 2020*). Air pollution levels have dropped significantly in India due to a massive dip in vehicular movement and industrial activity which have resulted in clean and fresh air (*Gautam, 2020*; *Mahato, Pal & Ghosh, 2020*; *Sharma et al., 2020*).

In the meantime, Kerala state government imposed a "triple-lockdown" in Kannur from 20 April for 20 days after identifying Kannur as a "hotspot" of Covid-19 and declared it a "red" zone in the state. The "triple-lockdown" involved a combination of technology and human surveillance and movement restrictions ple by providing one entry and exit point in all abodes leaving all other roads closed for traffic. People in red zones were not allowed to leave their houses and essential items were made available local authorities

through specific requests. Further, guards were deployed to ensure that people in the containment zones stayed indoors during the entire period of the lockdown. During the general lockdown and triple-lockdown periods, the skies over the polluted cities quickly cleared and smelled of fresh air. This offered a rare occasion for investigating how the air pollution levels responded to an extraordinary development.

In this work, we describe the trend in air pollution via monitoring the variations of surface $O_3$, NO and $NO_2$, $PM_{10}$ and $PM_{2.5}$, CO, VOC's including benzene, toluene, ethyl benzene, xylene, o-xylene (collectively called BTEX), $SO_2$, $NH_3$, and meteorological parameters at the time of pre-lockdown, lockdown and triple-lockdown days at Kannur town in the Kerala state of South India.

## METHODOLOGY

### Observational site

Kannur was the British military headquarters on the west coast of India until 1987. Kannur has had its industrial importance from very early days. It was an important trading center of the 12th century with an active business relationship with European and Arab countries. When the state of Kerala was formed in 1957, it was named Kannur town, because administrative offices were established in the district. Kannur is the sixth-most urbanized district in Kerala, with more than 50% of its residents living in urban areas. The district is known for its high level of literacy and health care. Kannur experiences a summer season from March to the end of May. This is followed by the south-west monsoon until September. Even the smallest pollution in the atmosphere of Kannur affects the quality of the air and subsequently the health of the people here. In the first phase of Covid-19, Kannur district reported the highest number of cases in Kerala till 15 May. Ground based observations were carried out at Kannur town to investigate the variations of different trace pollutants in the atmosphere from pre-lockdown days to triple-lockdown days. The observational site lies in a coastal belt along the Arabian Sea and is very close to the National Highway (NH 17). Observational site at Kannur town (11.87° N 75.37° E 3 m msl) in northern part of Kerala state. Kannur in south India is shown in the Fig. 1A and the aerial view of Kannur town and surroundings with observational site is shown in the Fig. 1B.

### Experimental setup

Observations of trace pollutants were carried out using the respective ground based gas analyzers from Environment S.A France. The measurements of the $O_3$ were made using a continuous $O_3$ analyzer (Model O342e) with a detection limit of 0.2.ppbv. Its working principle is based on $O_3$ detection by direct absorption in UV light. $O_3$ absorption spectrum is intense in the 250 and 270 nm wavelength range. Thus, it corresponds to the maximum range of $O_3$ absorption at 255 nm. NO, $NO_2$ and $NH_3$ were measured with the aid of gas analyzer (Model AC32e) with a detection limit of 0.2 ppbv. Its working principle is based on the NO chemiluminescence in the presence of highly oxidizing $O_3$ molecules. The NO in the ambient air is oxidized by $O_3$ to form excited $NO_2$ molecules. The concentrations of NO, $NO_2$ were measured based on the spectrum of the radiation

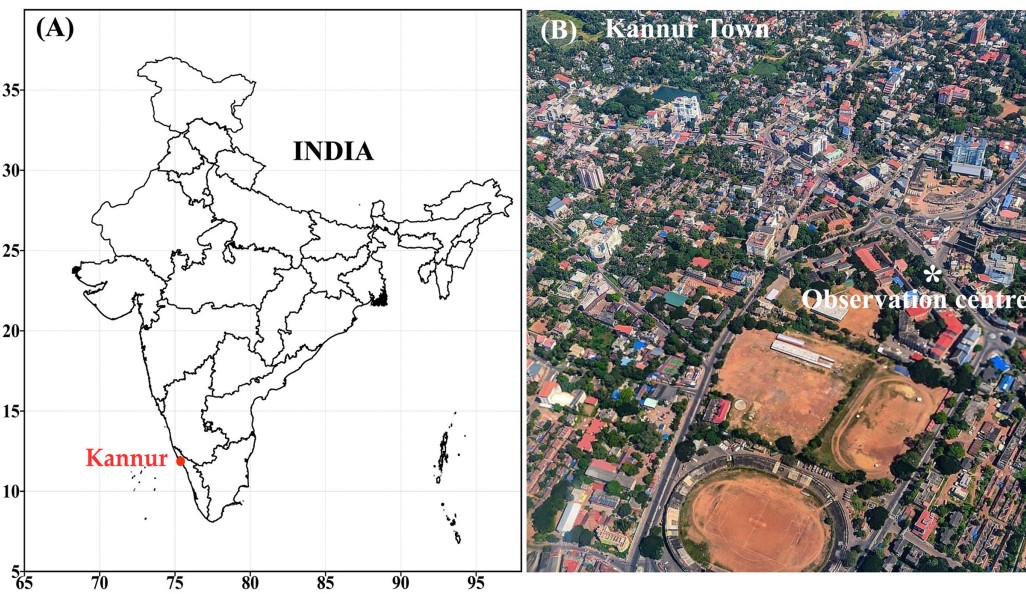

**Figure 1** (A) Kannur in South India (B) aerial view of Kannur town and the observational site.

emitted by $NO_2$ molecules at the excited level. Particulate matters ($PM_{10}$ and $PM_{2.5}$) were measured by using suspended particulate beta gauge monitor (Model MP101M).
Its working principle is based on the particle measurement by beta radiation attenuation.
The measurement consists of calculating the absorption difference between a blank filter and a loaded filter, knowing that the beta ray absorption follows an exponential law and is independent of the physiochemical nature of the particles.

Measurements of CO were made by using an analyzer (Model CO12e) with a detection limit of 0.05 ppm. Its working principle is based on CO detection by absorption in infrared light. VOC's (BTEX) were measured, based on gas chromatography coupled with a PID detector by using (VOC72e) analyzer. $SO_2$ measurements were made by using a UV fluorescent sulfur dioxide analyzer (Model AF22e) with a detection limit of 0.4 ppbv.
The ambient air to be analyzed is filtered by a hydrocarbon removing aromatic molecule device. The hydrocarbon molecule free sample to be analyzed is sent to a reaction chamber, to be irradiated by an UV radiation centered at 214 nm, which is the $SO_2$ molecule absorption wavelength. All the gas analyzers have been calibrated by using sample gases on a regular basis. The total solar radiation was measured by LSI LASTEM Italia (DPA870) pyranometer and surface air temperature measured by an external Pt100 sensor.

## Air Quality Index (AQI) calculation

The AQI is a scale designed to help us to understand the quality of the air we breathe and it also helps provide advice on how to improve air quality. Further, this index provides special awareness to the public who are sensitive to air pollution (*Beig, Ghude & Deshpande, 2010*). Normally air pollution levels may be higher in towns, near power plants and large stationary emission sources. To identify the overall improvement in air quality over Kannur, AQI was calculated and the details of AQI are available elsewhere

**Table 1 Breakpoint of different pollutants for AQI calculation, by *CPCB (2014)*.**

| AQI category (Range) | $PM_{10}$ 24-h | $PM_{2.5}$ 24-h | $NO_2$ 24-h | $SO_2$ 24-h | $NH_3$ 24-h | $O_3$ 8-h | CO 8-h |
|---|---|---|---|---|---|---|---|
| Good (0–50) | 0–50 | 0–30 | 0–40 | 0–40 | 0–200 | 0–50 | 0–1.0 |
| Satisfactory (51–100) | 51–100 | 31–60 | 41–80 | 41–80 | 201–400 | 51–100 | 1.1–2.0 |
| Moderate (101–200) | 101–250 | 61–90 | 81–180 | 81–380 | 401–800 | 101–168 | 2.1–10 |
| Poor (201–300) | 251–350 | 91–120 | 181–280 | 381–800 | 801–1200 | 169–208 | 10.1–17 |
| Very poor (301–400) | 351–430 | 121–250 | 281–400 | 801–1600 | 1,200–1,800 | 209–748 | 17.1–34 |
| Severe (401–500) | 430+ | 250+ | 400+ | 1,600+ | 1,800+ | 748+ | 34+ |

(*CPCB, 2014*; *Sharma et al., 2020*). The AQI is divided into five categories: good (0–50), satisfactory (51–100), moderate (101–200), poor (201–300), very poor (301–400) and severe (401–500) respectively. The observed concentrations of $PM_{10}$, $PM_{2.5}$, $NO_2$, $SO_2$, $O_3$, CO and $NH_3$ were converted into AQI using standard value. The AQI for each pollutant was calculated by the following formula given by *Sahu & Kota (2017)*.

$$AQIi = \frac{I_{HI} - I_{LO}}{Break_{HI} - Break_{LO}} \times (C_i - Break_{LO}) + I_{LO}$$

Where $C_i$ is the observed concentration of the pollutant "i"; $Break_{HI}$ and $Break_{LO}$ are breakpoint concentrations greater and smaller to $C_i$; and $I_{HI}$ and $I_{LO}$ are corresponding AQI ranges. Breakpoint concentration of different pollutants are provided by *CPCB (2014)* and is shown in Table 1.

# RESULTS AND DISCUSSION

## Diurnal variation of surface $O_3$, NO, $NO_2$

In order to study the impact of lockdown on the variation of trace pollutants over Kannur, the study period was divided into three spans; namely pre-lockdown period of 32 days (1–24 March 2020 and 10–17 May 2020), lockdown period of 25 days (25 March–19 April 2020), and a triple-lockdown period of 20 days (20 April–9 May 2020). $O_3$ concentration at Kannur town was very low (8–20 ppbv) at night time and high (15–55 ppbv) during the day time.

Figure 2A depict a diurnal variation of $O_3$ in Kannur town, and it shows that $O_3$ was observed to be high during afternoon hours due to the photolysis $NO_2$ in the presence of VOC's, CO and $CH_4$. The observed low concentration at night-time was mainly due to the loss of $O_3$ by the titration with NO (*Nishanth et al., 2014*). Diurnal variation of $O_3$ showed a similar pattern in pre-lockdown days, lockdown days, and triple-lockdown days but with differences in their concentrations. The maximum concentration of $O_3$ observed on pre-lockdown 4.74 (ppbv), lockdown days and triple-lockdown days were (45.64 ± 4.74 days), (49.58 ± 3.1 ppbv) and (55.66 ± 2.61 ppbv) respectively. Thus, an enhancement in $O_3$ concentration was observed over Kannur from pre-lockdown days to triple-lockdown and this increase is 22%.

The diurnal variations of NO and $NO_2$ are shown in the Figs. 2B and 2C respectively. NO concentration was observed high in night and early morning hours and found to

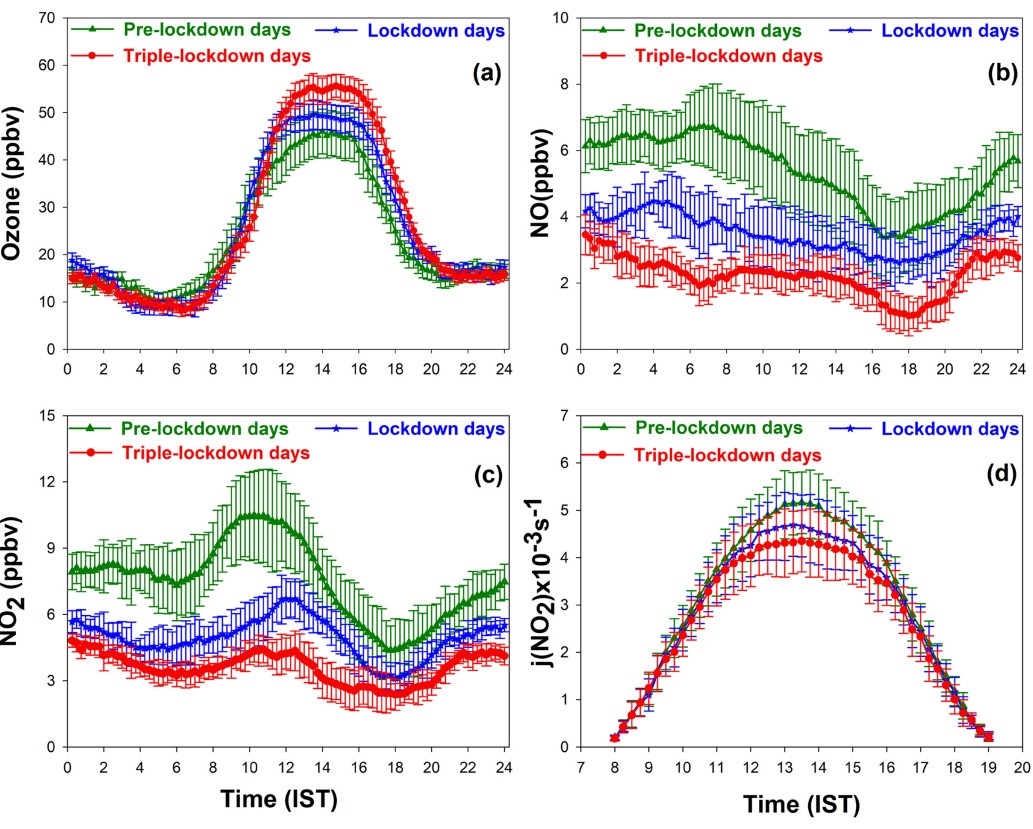

**Figure 2** Diurnal variation of (A) surface ozone (B) NO, (C) NO$_2$, (D) j(NO$_2$) between 1 March and 17 May, 2020.

be low during afternoon hours. The diurnal average concentrations of NO observed during pre-lockdown, lockdown and triple-lockdown days were (5.4 ± 1.2 ppbv), (3.5 ± 1.1 ppbv) and (2.1 ± 0.82 ppbv) respectively. Thus, the concentration of NO was found to be declined considerably from pre-lockdown days to triple-lockdown days, and the decrease was 61%. The domination of O$_3$ titration in the presence of high concentration of NO is the primary reason for the observed low concentration of O$_3$ during pre-lockdown days.

In pre-lockdown days, NO$_2$ concentration was found to increase in daytime due to enhanced emissions from vehicles and industries. The diurnal average concentrations of NO$_2$ observed during pre-lockdown, lockdown and triple-lockdown days were (9.6 ± 2.1 ppbv), (4.9 ± 1.8 ppbv), and (2.8 ± 0.88 ppbv) respectively. Hence, the concentration of NO$_2$ was found to be decreased significantly from pre-lockdown days to triple-lockdown days and the observed decrease was 71%. Conversely, O$_3$ concentrations observed were higher on triple-lockdown and lockdown period than pre-lockdown period, even in the absence of industrial activities and low traffic. Certainly, relatively lesser release of NO during lockdown and triple-lockdown days reduces the O$_3$ scavenging, and hence improved the photochemical production of O$_3$ from its other precursors.

At city-scale, VOC-NOx ratio is the key factor of O$_3$ formation (*Pusede & Cohen, 2012*). The urban areas are characterized by a low value of this ratio due to high NOx concentrations (*Beekmann & Vautard, 2010*). In an environment with "VOC-limited"

conditions, VOCs concentration is highly sensitive in $O_3$ formation in an environment with high NOx emission. Likewise, the ratio of $(NO_2)/(NO)$ that depends on the local concentration of $O_3$, since it is produced by the photodissociation of $NO_2$ and its sink is titration with NO. Thus, $O_3$ increase is due to a lower titration of $O_3$ by NO due to the strong reduction in local NOx emissions by road transport (*Sicard et al., 2020*). However, the presence of VOC's and NOx allows the formation of $O_3$ through $NO_2$ photolysis through a complex chemistry (*Monks et al., 2015*).

The day time average concentrations of $(NO_2/NO)$ during pre-lockdown, lockdown and triple-lockdown days were estimated to be (1.82 ± 0.8 ppbv), (1.38 ± 0.6 ppbv) and (0.86 ± 0.5 ppbv) which indicated a reduction of $(NO_2/NO)$ by 47% from pre-lockdown days to triple-lockdown days. The ratio between average concentrations of BTEX and NOx were found to be (1.2 ± 1.6 ppbv), (1.65 ± 1.1 ppbv) and (1.96 ± 0.82 ppbv) during pre-lockdown, lockdown and triple-lockdown days respectively; with an increase of 63% from pre-lockdown days to triple-lockdown. During the lockdown period, $O_3$ lapse rate due to the titration of NO might be less than its photochemical production from its precursors, and this may be the primary reason for the enhancement in $O_3$ observed. Further, a strong possibility of VOCs emission from home (e.g., cleaning fireplaces, painting) and garden activities (e.g., biomass burning) may also have contributed to the $O_3$ increase (*Su, Mukherjee & Batterman, 2003*; *Murphy et al., 2007*; *Wolff, Kahlbaum & Heuss, 2013*) in triple-lockdown days. Thus, the reduced concentration in NOx, and any enhancement of biogenic VOC's and their transport may have played a promising role in the enhancement of $O_3$ during triple-lockdown in Kannur town like other cities, which could be confirmed only after further investigations.

Diurnal variation of photo-dissociation rate coefficient $j(NO_2)$ was computed to estimate the strong dependance of $NO_2$ on the observed enhancement of $O_3$, during lockdown and triple-lockdown days as shown in Fig. 2D. The $j(NO_2)$ values exhibited the typical pattern of increasing gradually after sunrise, attaining a maximum value during noontime and decreasing during evening-time. The values of $j(NO_2)$ gradually increasing in tune with the intensity of solar radiation reaching on the surface. During pre-lockdown days, $O_3$ concentration was increasing in full harmony with the variation of $j(NO_2)$. Therefore, a positive correlation is observed between $O_3$ and $j(NO_2)$ during day time hours and the photo-dissociation coefficients were low due to the reduced concentration of $NO_2$ in lockdown and triple-lockdown periods. It was further observed that $j(NO_2)$ values were decreased in pre-lockdown days and triple-lockdown days by a percentage of 16%. This reveals that the $O_3$ production was the result of photo-dissociation of $NO_2$ in the presence of biogenic VOCs during lockdown and triple-lockdown days.

## Diurnal variation of particulate matters ($PM_{10}$ and $PM_{2.5}$)

The diurnal variations $PM_{10}$ and $PM_{2.5}$ during the study period are shown in the Figs. 3A and 3B. During pre-lockdown days $PM_{10}$ and $PM_{2.5}$ showed two peaks; of which one peak in the morning (07:00–10:00) hours and the other one in the late evening to night time (19:00–22:00) hours. Further, moderate levels were observed from night till the early morning hours due to shallow boundary layer (*Yadav et al., 2014*; *Qu et al., 2018*).

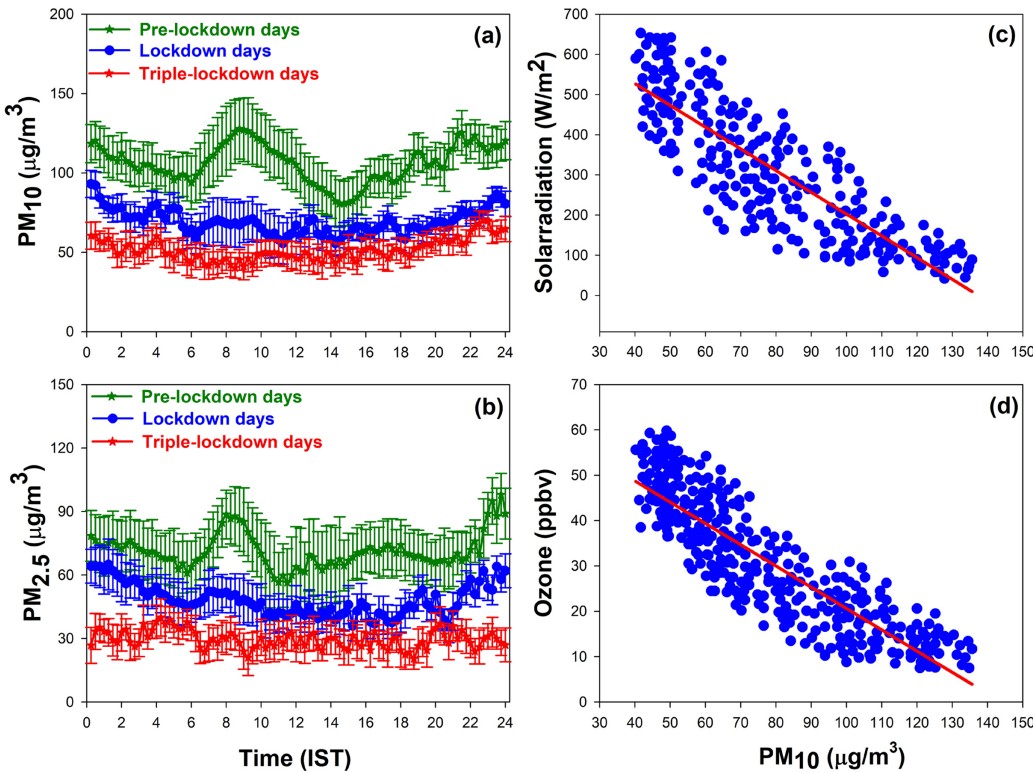

**Figure 3** Diurnal variation of (A) $PM_{10}$ (B) $PM_{2.5}$ and the scatter plot showing the linear correlation between (C) $PM_{10}$ and solar radiation (D) $PM_{10}$ and surface $O_3$.

Concentrations of particulate matters showed a morning peak followed by a decline in the afternoon on pre-lockdown days. The observed low concentrations of particulate matters during the afternoon hours can be attributed primarily to the dilution of particles linked with broaden boundary layer and also lesser traffic (*Stafoggia et al., 2019*).

In the lockdown days, the vehicular emissions were considerably reduced, and the observed magnitude of the morning peak was fairly small; whereas in triple-lockdown days the peak was absent due to the roads were deserted. The diurnal average concentration of $PM_{10}$ observed on pre-lockdown days, lockdown days and triple-lockdown days were ($127.8 \pm 21$ μg/m$^3$), ($70.96 \pm 12.6$ μg/m$^3$) and ($50.2 \pm 10.11$ μg/m$^3$) respectively. The concentration of $PM_{10}$ was found to decrease significantly from pre-lockdown days to lockdown days (45%) and lockdown days to triple-lockdown days (29%). Likewise, the diurnal average concentration of $PM_{2.5}$ on pre-lockdown days, lockdown days and triple-lockdown days were ($69.4 \pm 17$ μg/m$^3$), ($45.5 \pm 8$ μg/m$^3$) and ($32.5 \pm 7.5$ μg/m$^3$) respectively. The concentration of $PM_{2.5}$ was found to be decrease extensively from pre-lockdown days to lockdown days (34%) and lockdown days to triple-lockdown days (29%).

Linear negative correlations are obtained between $PM_{10}$ and solar radiation (Fig. 3C) and surface $O_3$ (Fig. 3D). The higher concentration of particulate matters on pre-lockdown days can reduce the photolysis rate $j(NO_2)$ in the lower troposphere and it can decrease photochemical production of $O_3$ on these days. During pre-lockdown days, the observed low

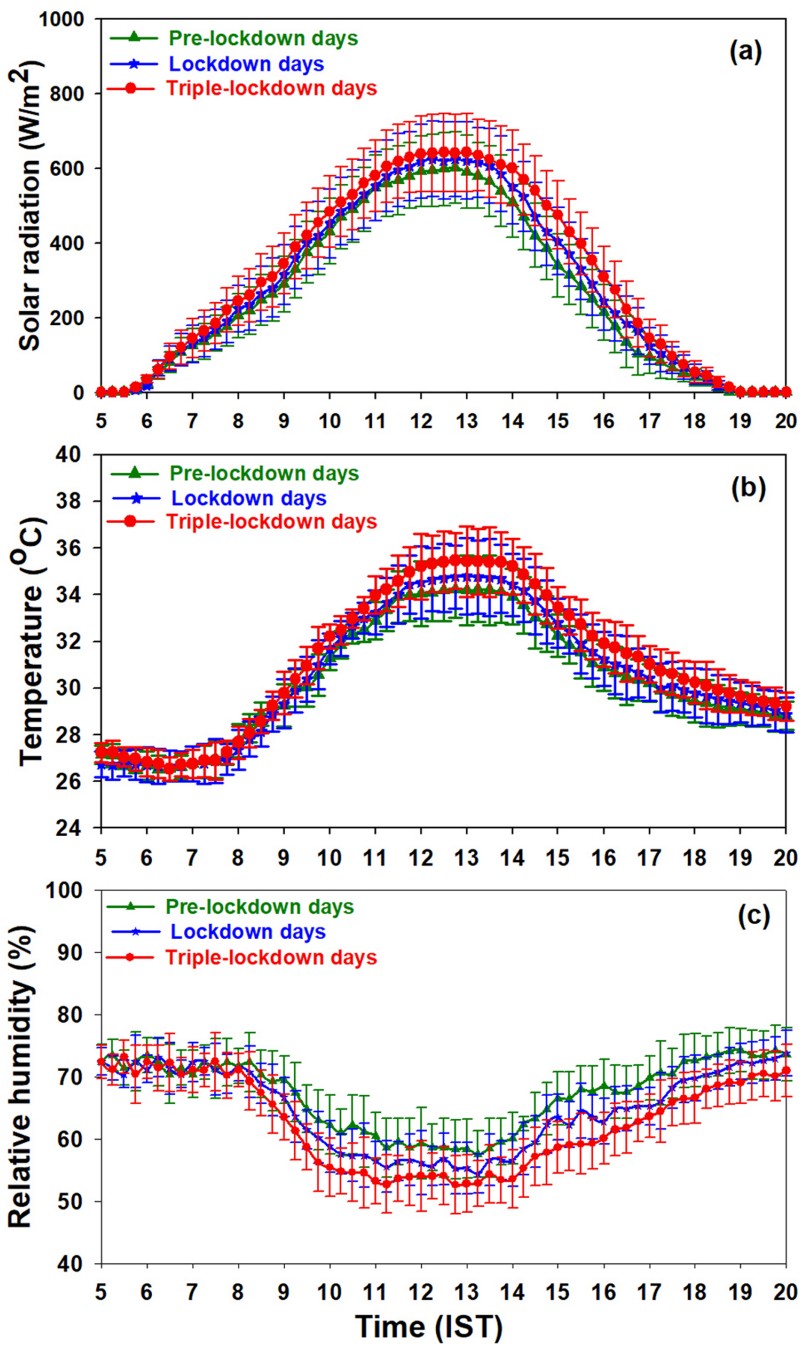

**Figure 4 Diurnal variation of (A) solar radiation (B) air temperature (C) relative humidity.**

concentration of $O_3$ indicates the different impacts of particulate matters on photolysis frequencies. A strong negative correlation coefficient (−0.91) between $PM_{10}$ and solar radiation reveals the active daytime photochemistry over Kannur town.

Diurnal variation of solar radiation, air temperature, and relative humidity observed over Kannur during the study period is shown in the Figs. 4A–4C respectively. Intensity of solar radiation and temperature were increase by 7% and 4% from pre-lockdown days

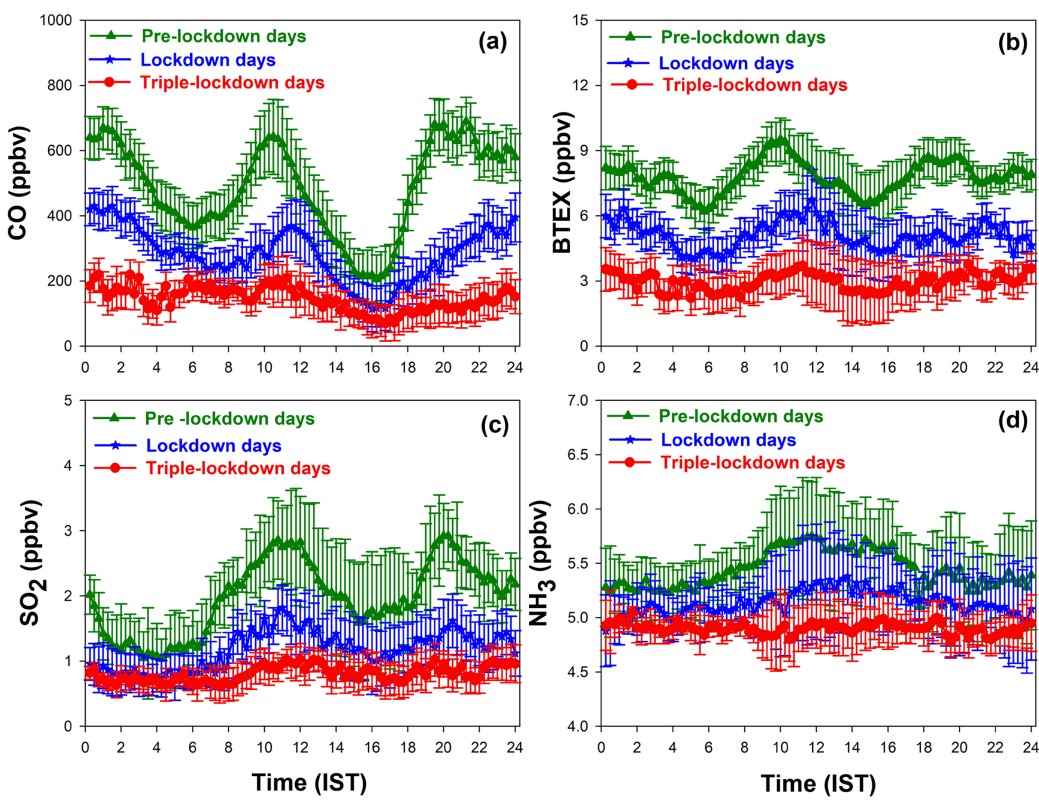

**Figure 5** Diurnal variation of (A) CO, (B) BTEX (C) SO$_2$ (D) NH$_3$ observed during the period.

to triple-lockdown days and relative humidity decreased by 8%. The increase in solar flux is attributed to the decline in the concentration of particulate matters (*Li et al., 2011*) in lockdown and triple-lockdown days, and this enhanced atmospheric temperature and declined relative humidity at this site.

## Diurnal variation of CO, BTEX, SO$_2$ and NH$_3$

Diurnal variation of CO, BTEX, SO$_2$ and NH$_3$ observed during the observational period are shown in the Figs. 5A–5D respectively. In pre-lockdown days, the diurnal variation of CO, BTEX and SO$_2$ shows two distinct peaks; one peak during morning (08:00–11:00) hours and other at late evening. The small duration of the morning peak was due to the expansion of boundary layer height whereas the large evening peak due to the shallow boundary layer. These peaks were associated with high traffic during the morning and evening hours. During lockdown days, a small peak observed in the morning was due to the presence of few vehicles; while the peak was absent in triple-lockdown days due to the roads were deserted.

The maximum concentration of CO observed in day time on pre-lockdown days, lockdown days, and triple-lockdown days were (642 ± 115 ppbv), (368 ± 79 ppbv) and (210 ± 70 ppbv) respectively. The concentration of CO was found to decrease significantly from pre-lockdown days to lockdown days, and lockdown days to triple-lockdown days. The maximum concentration of BTEX observed in day time on pre-lockdown days,

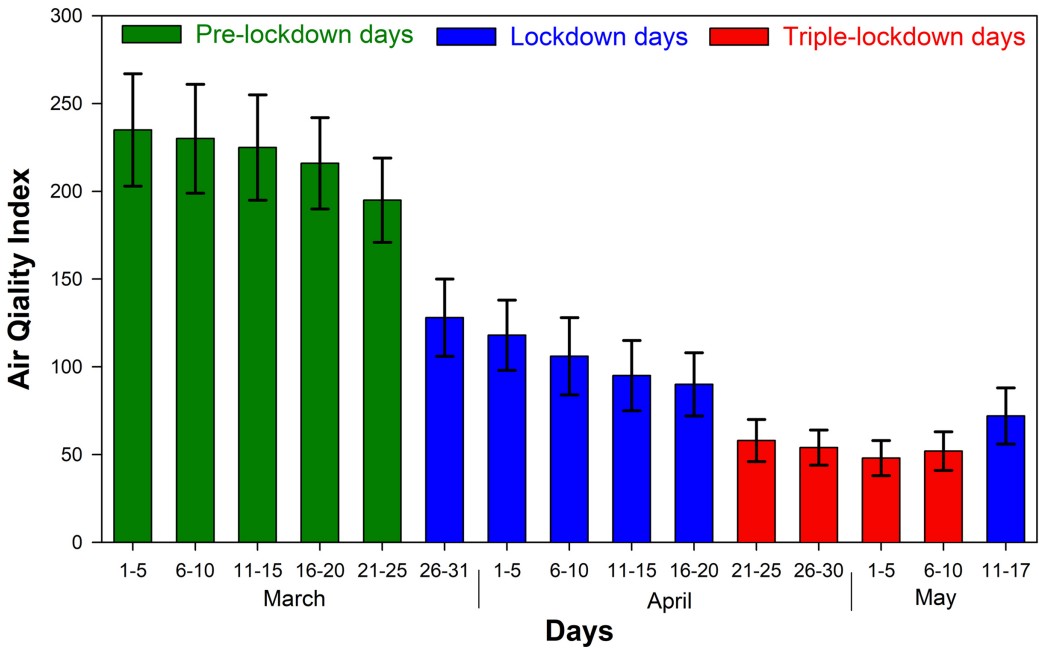

**Figure 6 Air quality index for pre-lockdown, lockdown and triple lockdown period.**

lockdown days and triple-lockdown days were (9.5 ± 1.4 ppbv), (6.7 ± 1.1 ppbv) and (3.7 ± 0.8 ppbv) respectively. The concentration of BTEX was found to be decreased from pre-lockdown days to lockdown days, and lockdown days to triple-lockdown days due to absence of vehicular and industrial emissions.

The diurnal variation of $SO_2$ was most pronounced during traffic hours over Kannur town. The maximum concentration of $SO_2$ observed in day time on pre-lockdown days, lockdown days, and triple-lockdown days were (2.86 ± 0.72 ppbv), (1.82 ± 0.45 ppbv) and (1.08 ± 0.32 ppbv) respectively. Thus the concentration of $SO_2$ was found to be decrease considerably from pre-lockdown days to lockdown days, and lockdown days to triple-lockdown days. During pre-lockdown period, the daily averaged $NH_3$ concentrations varied from 5.1 to 5.8 ppbv. Daily $NH_3$ exhibited a temporal variation with higher concentrations on the noon time hours; due to higher air temperatures and lower wind speeds. High air temperatures will favor $NH_3$ volatilization, and the low wind speeds support the accumulation of air pollutants (*Wang et al., 2015*; *Zhao et al., 2016*). The maximum concentration of $NH_3$ observed in day time on pre-lockdown days, lockdown days, and triple-lockdown days were (5.84 ± 0.52 ppbv), (5.32 ± 0.41 ppbv) and (4.91 ± 0.32 ppbv) respectively. Like other trace pollutants, the concentration of $NH_3$ was also found to be decreased from pre-lockdown period to triple-lockdown days due to complete shutdown of traffic and industrial activities in Kannur district.

## AQI during the study period

The overall AQI of a day is the maximum AQI of the constituent pollutant, and the corresponding pollutant is the dominating pollutant. Figure 6 represents the variation of

AQI during the study period over Kannur town. The highest index of 235 ± 32 which refers to poor air quality was observed in pre-lockdown period from 1 to 5 in March, during which the vehicular traffic was quite normal. From the figure, it is observed that the AQI was declining from 25th March coincides with the beginning of lockdown. In the first 5 days of the lockdown its average value was changed to 128 ± 22 and then to 118 ± 20 in the following days. Since the triple-lockdown in Kannur was implemented on 20th April, the value of AQI further come down and reached its lowest level at 48 ± 10 due to the ban of complete vehicles on the road. As a result, air pollution was highly reduced by which the air became unpolluted at this site. Thus, the enhanced air quality resulted from the triple-lockdown may prevent the virus spread in Kannur district.

## CONCLUSION

The present study revealed a drastic reduction in air pollution over Kannur a town in north of Kerala state India, which was identified as one of the "first hotspots" of COVID 19 by the Government of India. Subsequently, a triple-lockdown was implemented from 20 April for further twenty days that restricted the movement of people. A considerable reduction in atmospheric air pollutants over this region was observed during the lockdown and triple-lockdown periods at this site. The highlights of the observation are the following:

- Surface $O_3$ concentration was increased to 22% in triple-lockdown period while NO and $NO_2$ concentrations were decreased to 61% and 71% from pre-lockdown days to triple-lockdown days. The primary reason for the increase in $O_3$ is due to the reduction in titration of $O_3$ with NO. The sensitivity of VOC's (both biogenic and industrial) could be confirmed by measuring them on a long-term basis. The calculated diurnal profile of $j(NO_2)$ values showed a maximum reduction in triple-lockdown period due to the decrease in $NO_2$.

- The concentration of CO, VOC's (BTEX), $SO_2$ and $NH_3$ were declined to 67%, 61%, 62% and 16% respectively from pre-lockdown days to triple-lockdown days.

- The concentration of $PM_{10}$ and $PM_{2.5}$ were decreased to 61% and 53% respectively. Further, the increase in solar flux observed was mainly due to the absence of scattering and absorption of aerosols present in the atmosphere due to lockdown.

- The AQI analysis revealed that the air quality in Kannur was quite improved during lockdown period.

The dramatic decline in air pollution during this lockdown has significant immediate consequences. Exposure to high levels of particulate matters and trace gases has substantial detrimental effects on human health. Many researchers have hypothesized that the drop in air pollution levels may currently be saving a significant amount of lives, not only by reducing individuals' susceptibility to COVID-19, but also by preventing world's seven million annual deaths due to air pollution exposure.

Yao et al. (2020) analyzed the death rate due to COVID-19 and spatial correlation of $PM_{10}$ and $PM_{2.5}$ in China, and they revealed that higher concentration of $PM_{10}$ and

$PM_{2.5}$ had a positive correlation with deaths caused by COVID-19. *Wu et al. (2020)* suggested that, long-term exposure to fine particulate matter ($PM_{2.5}$) induced a risk of COVID-19. Further, they found that an increase of 1 $\mu g/m^3$ in $PM_{2.5}$ enhanced 8% morbidity due to COVID-19 in the United States. *Fattorini & Regoli (2020)* suggested that, chronic exposure of particulate matters ($PM_{10}$ and $PM_{2.5}$) could aggravate the susceptibility of COVID-19 infection. A positive correlation exhibited between concentrations of $PM_{10}$, $PM_{2.5}$, $SO_2$, $NO_2$, CO and COVID-19 outbreak in California (*Bashir et al., 2020*). It is reported, that air pollution has a positive association with COVID-19 confirmed cases in China (*Zhu et al., 2020*). Thus, this improvement in air quality during lock down might have played a vital role in bringing reduced susceptibility of virus getting infected into the lungs. The enhancement in the air quality induced by triple-lockdown might be one of the prime reasons by which Kerala could flatten the transmission curve for the first time in India. This association of air quality and spread of COVID if confirmed by future studies suggest that air quality should also be considered as part of an integrated approach towards human health protection and prevention of the epidemic spreads.

## ACKNOWLEDGEMENTS

The authors wish to thank Dr. Shyam Lal Program Director of (AT-CTM) for his support and inspiration. Resmi expresses her gratitude to Dr. R. Venkatachalam (Principal) and Dr. D. Manivannan (HOD of Physics) of Erode Arts and Science College Tamil Nadu for providing the necessary facilities. Authors gratefully acknowledge the effort made by Dr. Gufran Beig, Academic Editor of Peerj, and the other three anonymous reviewers for the critical reviewing of the manuscript and making valuable suggestions for the improvement of the manuscript in the present form.

### Funding

This work was carried out with the support of ISRO- GBP (AT-CTM) program and Kerala State Council for Science Technology and Environment (KSCSTE) and the project sanction order no. Council (P) Order No. 186/2009/KSCSTE dated 17.2.2009. KT Valsaraj received support from the Charles and Hilda Roddey Distinguished Professorship in Chemical Engineering at LSU. The funders had no role in study design, data collection and analysis, decision to publish, or preparation of the manuscript.

### Grant Disclosures

The following grant information was disclosed by the authors:
ISRO-GBP (AT-CTM).
Kerala State Council for Science Technology and Environment (KSCSTE): 186/2009/KSCSTE.
Louisiana State University (LSU).

## Competing Interests

The authors declare that they have no competing interests.

## Author Contributions

- C.T. Resmi performed the experiments, prepared figures and/or tables, and approved the final draft.
- T. Nishanth conceived and designed the experiments, performed the experiments, analyzed the data, prepared figures and/or tables, authored or reviewed drafts of the paper, and approved the final draft.
- M.K. Satheesh Kumar conceived and designed the experiments, performed the experiments, analyzed the data, prepared figures and/or tables, authored or reviewed drafts of the paper, and approved the final draft.
- M.G. Manoj analyzed the data, prepared figures and/or tables, and approved the final draft.
- M. Balachandramohan performed the experiments, analyzed the data, prepared figures and/or tables, and approved the final draft.
- K.T. Valsaraj conceived and designed the experiments, analyzed the data, authored or reviewed drafts of the paper, and approved the final draft.

## Field Study Permissions

The following information was supplied relating to field study approvals (i.e., approving body and any reference numbers):

The observations are carried out with the support of ISRO-GBP(AT-CTN) project and Kerala state council of science, Technology, Environment and it is a long term continuous program. The approval reference number is Council (P)Order No. 186/2009/KSCSTE (dated 17.2.2009). The observations were carried out Kannur Town, and permission to access the private building where sampling took place was provided by Rama Chandra.

## Data Availability

The raw data is available as a Supplemental File.

## Supplemental Information

Supplemental information for this article can be found online at http://dx.doi.org/10.7717/peerj.9642#supplemental-information.

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
