# Peer review of "Air quality improvement during triple-lockdown in the coastal city of Kannur, Kerala to combat Covid-19 transmission"

_PeerJ, doi:10.7717/peerj.9642_

## Round 0.1 · original submission · Minor Revisions

· Academic Editor

Minor Revisions

Reviewers have raised a number of suggestions. Please incorporate all of them and respond to each comment in order.

Reviewer 1 ·

Basic reporting

No comment

Experimental design

No comment

Validity of the findings

No comment

Additional comments

Air quality improvement during triple lockdown in the coastal city of Kannur, Kerala to combat COVID-19 transmission
The authors have presented the analysis of the data of atmospheric trace gases, particulate matter, meteorological parameters collected at a site in Kannur, Kerala, during the 54 day phased lockdown imposed by the Government of India to prevent the spread of COVID-19 pandemic. All anthropogenic activities including transportation had come to a near halt during this period across India. As expected, there is considerable reduction in the concentrations of trace gases as well as particulate matter at the observational site at Kannur. It is interesting to know the effect of the reduced emissions on ozone chemistry. The ozone production depends on NO2/NO ratio and VOC/NOx ratio. The authors report 22 % increase in ozone production during triple lockdown. This is attributed to increase in VOC/NOx ratio in a limited VOC environment. However, NO2/NO ratio is not discussed. Reduced NO concentration reduces the titration of ozone. The ozone production is also influenced by the increase in solar radiation at ground level due to decrease in the aerosol concentrations. However, the role of reduced CO emission on ozone formation is not discussed in the present paper. Even though a rigorous quantitative analysis is not presented, the paper need to be considered for publication as the first information about the air quality under a special situation of the lockdown. However, following modifications/clarifications are suggested/sought:
In the document that I have downloaded, the commas are missing in many places in the entire paper.
The word lockdown or lock down-there is no uniformity.
English needs to be corrected: Few examples are:
1. Line no. 50:…………….were found to decrease……….(instead of) were found to be decrease………..
2. Line no. 64: …………caused a severe downturn…………….instead of severely downturn………..etc.
3. Line no. 102: ……..northern part of Kerala state…….(instead of) north of Kerala state…
4. Line no. 152: …..and hence enhanced the……..(instead of) hence improved the
Line nos. 143-145: ‘’NO2 was found to increase in daytime due to the enhanced emission of hydrocarbons from vehicles and industries and their photochemical reactions to produce O3 between 08:00 and 12:00 hours in pre-lockdown days.’’-How is NO2 increase related to enhanced emission of hydrocarbons?
Line nos. 157-159: How to claim that NOx emitted is less than VOC? Alternately, somewhere in the paper it is better to mention that the biogenic part of VOC emissions are likely to be unaffected by the lockdown. As mentioned in line nos. 159-161, there is a possibility of increase in the emission of biogenic VOCs due to related human activities. The authors may slightly elaborate on this.
Line nos. 174-175: This is also due to reduced O3 titration under reduced NO emissions. This aspect also may be mentioned.
General comment: During the triple lockdown, due to reduction in the aerosol concentrations, the solar radiation reaching the surface has increased leading to increase in the surface temperature and decrease in the humidity. CO emissions have also reduced considerably. Under this scenario, the OH/HO2 radical production is modified. What is the role of this in O3 production?
Line nos. 196-197: The sentence sits better in the previous paragraph.
Line nos. 222-224: “These peaks were associated with high traffic during the morning hours. During lockdown days the small peak observed in the morning was due to the absence of vehicles while the peak was absent in triple lockdown days due to the roads were deserted.” The sentence is not clear-is it-the presence of few vehicles?
Line nos. 229-239: BTEX being emitted by the industrial activities obviously shows decrease during the lockdown. The O3 increase during the triple lockdown is attributed by the authors to the biogenic VOCs. As suggested in the comment above, the authors need to mention that the biogenic VOC emissions are in principle not affected during the lockdown. On the contrary, there is a possibility that they have increased due to the related human activities as mentioned in the lines 159-161.
Line nos. 279-281: This conclusions may be further diluted.
Line nos. 283-286: The generalization of the results on the national level is not appropriate here. Either these sentences be dropped or moved with suitable modifications to the end after drawing conclusions of the present study.

Reviewer 2 ·

Basic reporting

This is an interesting paper.
Mainly data has been reported

Experimental design

1.Detail location of observations(name of site) conducted may be given

2. How frequently calibration of instruments are done

Validity of the findings

I presume the AQI is calculated as per Nasir and Brahmaich (2015). Since authors are discussing indian data then it should be calculated as per CPCB standards with reference. This may be clarified.

Additional comments

Since there is a limited data to date and to suggest there may be a positive association between long term exposure to ambient air pollutin and COVID-19 mortality.
The line 301 to 306 may be discussed more.
The restrictions have caused a considerable fall in level of air pollution.Thus preventing many avoidable deaths from non COVID-19 causes.

Reviewer 3 ·

Basic reporting

No comment

Experimental design

No comment

Validity of the findings

No comment

Additional comments

Manuscript number (#49486)
Title: Air quality improvement during triple lockdown in the coastal city of Kannur, Kerala to combat COVID-19 transmission

General comment
This current study discussed the impact of COVID-19 lockdown on trace gases (O3, CO, NOx, BTEX and NH3) and particulate matter (PM2.5 and PM10) at Kannur city in the Kerala state of South India. The air pollution data have been analyzed along with meteorological parameters during the pre-lockdown, lockdown and triple lockdown period to see the change in the pollution levels for lockdown periods. The study highlights the mixing ratios of surface ozone were increased by ~22% during lockdown period by comparing the pre-lockdown period. NO and NO2 concentrations were decreased by 90% and 144% respectively. PM10 and PM2.5 were declined significantly by 61% and 53% respectively. Additionally, the concentration of CO, VOCs, SO2, and NH3 were found to be decrease significantly from pre-lockdown days to triple lockdown days. Overall, the study explains the air quality was improved during lockdown periods at the study site.
Authors has appropriately presented their results and discussed comprehensively their findings. Hence, this paper is suitable for the publication in the present journal after the following comments / suggestions are incorporated.
Comments/Suggestions
In the Abstract:
More quantitative information should be included. No need to mention 34-36 lines in abstract section so please move these lines in introduction section.
Line 39-41: Please rewrite the sentence “trend in air pollution via monitoring the variations of surface Ozone Oxides of Nitrogen Carbon Monoxide Sulphur Dioxide Ammonia Volatile Organic Compounds Particulate Matters (PM10 and PM2.5) and meteorological parameters at the time of pre-lockdown lockdown and triple lockdown days at Kannur town in the Kerala state of South India.” and also use the comma after each species (for example: surface ozone, oxides of nitrogen…..)
Line 43: there is typo error. Use comma after “From pre-lockdown days to triple lockdown days” and also O from Ozone should be write in small letter.
Line 45: Please clearly explain about 144%.
Line 50: “The concentration of CO VOCs SO2 and NH3 were found to be decrease significantly”
Authors should be included the reduction values at least for few species (CO and SO2)

Keywords: Authors may change the trace pollutants by air pollutants (if possible)

In Introduction:
The introductory part needs to be redrafted. Few lines about the pollutants should be included. For example: sources and their roles. Specifically, BTEX role on health
Authors may cite latest references (for sources and their roles….. etc)

Liu et al., (2008), Volatile organic compound (VOC) measurements in the pearl river delta (PRD) region, China. Atmos. Chem. Phys., 8 (2008), pp. 1531-1545, 10.5194/acp-8-1531-2008
Yadav et al., 2016. Role of long-range transport and local meteorology in seasonal variation of surface ozone and its precursors at an urban site in India. Atmospheric Research, 176-177, 96-107.
Yadav et al., 2019. The role of local meteorology on ambient particulate and gaseous species at an urban site of western India. Urban climate, 28, 100449.

Line 95-99: Rewrite the plan. First use full name of species in the introduction then use their abbreviations properly. Here some are abbreviated (surface O3 NO NO2 CO SO2 NH3 Volatile Organic compounds) which makes confusion.

In Observational site and experimental setup:

Please write 2-3 lines about the working details of each instrument.
Results and Discussion:
Line 123: Authors may use latest reference along with Fishman and Crutzen 1978
For example:
Maji, et al., (2020), Winter VOCs and OVOCs measured with PTR-MS at an urban site of India: Role of emission sources, meteorology and photochemical sources. Environmental Pollution, 258, 113651.
Yadav, et al., (2019), Investigation of emission characteristics of NMVOCs over urban site of Western India. Environmental Pollution, (2252), 245-255,

Authors should be included only outcomes and their discussion in this section. The common part “Particulate matters (PM10 and PM2.5) are complex mixture of organic and inorganic substances found in the ambient air and they play a vital role on the radiation budget of the atmosphere via the scattering and absorption processes (Qu et al. 2017; Qu et al. 2018).” should move in the introduction section.
At few places’ sentences are bit long and merged so these long sentences should be broken into small sentences which will be helpful for readers.

There are some grammatical errors, so paper should be thoroughly proof read

Though authors have discussed their findings but still I feel more discussion.

The summary and conclusion section should be point-wise for better readability.

Annotated reviews are not available for download in order to protect the identity of reviewers who chose to remain anonymous.

---

## Round 0.2 · accepted · Accept

· Academic Editor

Accept

Author has incorporated all changes as per comments.